# Development and validation of predictive workable weight equations for Brazilian older adult residents in long-term care institutions

**Marcos Felipe Silva de Lima**[1]*, **Natália Louise de Araújo Cabral**[1], **Larissa Praça de Oliveira**[1], **Laura Camila Pereira Liberalino**[1], **Severina Carla Vieira Cunha Lima**[2], **Lucia de Fátima Campos Pedrosa**[2], **Kenio Costa de Lima**[3], **Clélia de Oliveira Lyra**[2]

1 Postgraduate of Public Health, Federal University of Rio Grande do Norte, Natal, Brazil, 2 Department of Nutrition, Federal University of Rio Grande do Norte, Natal, Brazil, 3 Department of Dentistry, Federal University of Rio Grande do Norte, Natal, Brazil

* marcosfelipe@ymail.com

## Abstract

### Background

Weight measurement is important in the nutritional anthropometric monitoring of older adults. When this measurement is not possible, estimates may be used.

### Aim

Developing and validating weight predictive equations for older adult residents in long-term care institutions in Brazil.

### Subjects and methods

The sample comprised 393 older adult residents in long-term care institutions. Data were collected in two stages, with 315 older adults in the first and 78 in the second. We have measured the arm, calf, and waist circumferences, as well as the triceps and subscapular skinfold and knee height. Multiple linear regression was used to develop the equations, which were evaluated through the coefficient of determination, standard error of estimation, Akaike information criterion, intraclass correlation coefficient (ICC), and Bland-Altmann plot.

### Results

Five models with different anthropometric measurements were developed, (1) arm circumference as a discriminant variable (ICC: 0.842); (2) best statistical fit for men and women (ICC: 0.874) and its stratification by sex (3) (ICC: 0.876); (4) easy-to-perform measurement for men and women (ICC: 0.842) and its stratification by sex (5) (ICC: 0.828).

### Conclusion

Five models for estimating the weight of older adult residents in long-term care institutions were developed and validated. The choice to use the models should be based on the physical capacity of the older adults to be evaluated.

**Data Availability Statement:** All files are available from the OSF database. DOI 10.17605/OSF.IO/ G76Y5.

**Funding:** This work was supported by the Fundação de Apoio à Pesquisa do Estado do Rio Grande do Norte (Rio Grande do Norte Research Support Foundation) (FAPERN): Programa de Apoio a Núcleos Emergentes (Emerging Centers Support Program) – Edict PRONEM/FAPERN/ CNPq 003/2011. In addition, it was financed in part by the Coordenação de Aperfeiçoamento de Pessoal de Nível Superior - Brasil (CAPES) - Finance Code 001. The funders had no role in study design, data collection and analysis, decision to publish, or preparation of the manuscript.

**Competing interests:** The authors have declared that no competing interests exist.

## Introduction

Weight is a measure that represents the sum of all body components of an individual in a given period and under certain conditions [1]. This measure may be influenced by the physiological state, fluid retention, limb loss, and food intake of individuals, or may vary as an adverse effect of certain medications [2]. Moreover, it is strongly associated with mortality outcomes for older adult residents of long-term care institutions [3–5].

Malnutrition, with consequent weight loss, is one of the factors most often associated with mortality among older adults [3–6]. Thus, offering adequate health care to the residents of Long-term Care Institutions for older adults (LTCIs) is important to monitor their nutritional status. Brazilian laws provide for monthly monitoring of the prevalence of malnutrition and involuntary weight loss in LTCIs to prevent health problems in this population [7].

Weight measurement is simple, requiring only that individuals stand up for a few seconds on a scale. However, for older adults with mobility restrictions, the procedure is more complicated, because the simple act of walking from one room to another can be complex and risky. Thus, the procedure itself may pose a risk to the physical integrity of older adults, raising the need for another weight assessment method [8, 9].

To support the assessment of nutritional status, studies have developed weight predictive equations for adults and older adults. Among them, Chumlea et al. (1988) is, remarkably, the most used work in clinical practice and epidemiological studies, as is that by Rabito et al. (2008), which was conducted with Brazilians [8, 9]. However, these studies have not been validated for older adult residents of LTCIs. In addition, the equations developed include anthropometric measurements—waist circumference, the subscapular skinfold, half arm span, and demi-span—that are difficult to perform in older adults who are bedridden and have severe joint stiffness [10].

Given the importance of applying anthropometric methods to detect possible changes in the nutritional status of older adults in LTCIs early, this study aims to develop and validate weight estimation equations for this group. Thus, we seek to provide subsidies to monitor nutritional status, perform future nutritional interventions, and make recommendations regarding methods for estimating body weight.

## Materials and methods

A cross-sectional study was conducted with older adults in LTCIs in Natal/RN, a city in northeastern Brazil, from October 2013 to June 2014. The study population comprised all older adult residents in LTCIs in this period. Those aged over 60 years who were present at the time of data collection were included in the study. Individuals with amputated limbs or any physical disability that prevented the obtaining of anthropometric measurements, such as the presence of sacral pressure ulcers and post-operative orthopedics, were excluded from the study.

Data were collected in two stages. In the first, which occurred between October 2013 and June 2014, data for the development of the equations were collected. The population universe of this study comprised 337 older adults, who lived in ten different LCTIs registered in the Municipal Health Surveillance Coordination. Of that number, 15 refused to participate and seven were not included because they were amputees, presented pressure ulcers, or postoperative orthopedics. Thus, the final sample comprised 315 patients of both sexes aged over 60 years based on the classification recommended by the World Health Organization [11], which is adopted in Brazil.

In stage 2, data were collected from December 2018 to March 2019 to validate the weight equations from four LTCIs in the metropolitan region of Natal/RN. An anthropometric evaluation was performed of older adult residents in the LTCIs. The eligibility criteria were the

same as those in stage 1. The number of residents at the beginning of the data collection was 138 older adults in four LTCIs, corresponding to the population universe. Of this total, 55 have also participated in the stage 1 of the research, and were excluded from stage 2. Four older adults refused to take part, and it was not possible to collect anthropometric data from one individual, resulting in 78 participants. For the anthropometric evaluation, calibrated equipment and two trained evaluators were used, both based on the Technical Measurement Error methodology (TEM) [12]. TEM values were:

a. Weight: 0.0 (intra-evaluator 1); 0.0 (intra-evaluator 2); 0.0 (inter-evaluator).

b. Height: 0.1 (intra-evaluator 1); 0.0 (intra-evaluator 2); 0.1 (inter-evaluator).

c. Arm circumference: 0.6 (intra-evaluator 1); 0.9 (intra-evaluator 2); 1.3 (inter-evaluator).

d. Waist circumference: 0.3 (intra-evaluator 1); 0.6 (intra-evaluator 2); 0.7 (inter-evaluator).

e. Calf circumference: 0.6 (intra-evaluator 1); 0.6 (intra-evaluator 2); 0.7 (inter-evaluator).

f. Triceps skinfold thickness: 2.0 (intra-evaluator 1); 1.6 (intra-evaluator 2); 4.6 (inter-evaluator).

g. Subscapular skinfold thickness: 1.8 (intra-evaluator 1); 2.0 (intra-evaluator 2); 6.2 (inter-evaluator).

h. Knee height: 0.5 (intra-evaluator 1); 0.5 (intra-evaluator 2); 0.7 (inter-evaluator).

Both evaluators were classified as skillful anthropometrists [12].

Registration forms were designed to collect information. For the anthropometric evaluation, we have measured the arm (AC), calf (CC), and waist (WC) circumferences; subscapular (SS) and triceps skinfold thickness (TS) and knee height (KH). The anthropometric measurement protocol followed the recommendations by Lohman et al. 1988 [13].

All measurements were evaluated in duplicate, starting with the upper limbs, trunk region, and then lower limbs. A Sanny® anthropometric measuring tape (São Paulo, Brazil); a scientific Lange® type plicometer (Michigan, USA), which is accurate to 1.0 mm; and a 100 cm anthropometer were used to measure the circumferences, skinfold thickness, and knee height, respectively. In addition to these measures, the body weight of the older adult participants who could walk was measured using a Balmak® electronic scale (São Paulo, Brazil) with capacity of 300 kg and precision of 50 g. The weight of those who were bedridden was obtained with a SECA® 985 scale (bed scale and electronic dialysis with trolley) (Bolton, England).

Data were analyzed using the IBM SPSS® software, version 22.0, for Microsoft Windows®. Descriptive data were presented. A Pearson correlation test was also performed to identify which anthropometric measure is more correlated with weight. This measure was used in a receiver operating characteristic (ROC) curve to find a cutoff point to differentiate the estimation models. Based on this cutoff point, two weight estimation equations were formulated.

To develop these equations, the sample was characterized and stratified by sex (male and female) and age group (60 to 74 years, 75 years or older). Thereafter, a multiple linear regression was performed to identify which combination of variables would be the most appropriate in estimating weight.

A multiple linear regression analysis was performed to select the most appropriate model to answer the study objective. The developed regression models consider statistical criteria of significance using the coefficient of determination ($R^2$), estimated standard error (SEE), and Akaike Information Criterion (AIC). SEE is an accuracy index, calculated by standard

deviation between repeated measures. The lower SEE obtained, the better the reliability. Among the tested models, the one including the measurements for arm and calf circumference and knee height was selected.

Furthermore, equations composed of measures easily obtained in older adults who are in a wheelchair or bedridden, i.e., calf circumference and knee height, were also formulated. This was based on Lima et al. (2016), who determined easy-to-measure variables in older adults with mobility restrictions. In addition, a Bland-Altmann plot was used to evaluate the model fit.

Finally, we have compared the characteristics of two groups (development and validation) by Student's t-test for independent samples as a requirement for validation. This was done by the intraclass correlation coefficient (ICC) and their confidence intervals (95% CI) with the observed and estimated weights using the protocols by Chumlea et al. (1988) and Rabito et al. (2008). The ICC was performed using a two-way mixed method. The test was of the absolute-agreement type, which is the ability to get the same value when a measurement is made of the same individual/population on a different occasion. The single-measure ICC was used because the estimate and observed value needed to agree, not their average. For all statistical tests, a significance value of 5% was adopted.

The Ethics Research Committee of the Federal University of Rio Grande do Norte approved the study under number 308/2012 (CAAE 0290.0.051.000–11) and 2.731.187/2018 (CAAE 84319418.5.0000.5292). The older adults, as well as those responsible for the LTCIs, received instructions regarding the research, and only those who signed the Informed Consent Form were included, as determined by the Brazilian National Health Council.

## Results

Most subjects were female (77.0%). There was a significant decrease in weight as age increased, with an average difference of 4.83 kg between the age groups. The average difference in weight in relation to the evaluated sex was greater. Men weighed 9.46 kg more than women at their average weight (Table 1).

The characterization of the study population showed that the anthropometric measurements were different for each studied group. Thus, we sought to verify which measure would best explain the variation in weight. Pearson's correlation analysis showed that the arm circumference was most correlated (r = 0.869) with the observed weight and was thus used to discriminate the older adults in relation to their weight.

The ROC curve indicated that an arm circumference measurement of 27 cm represented the best sensitivity and 1-specificity. In Table 2, it is shown the age and other anthropometric measurements for individuals grouped in the categories AC ≤ 27 cm and AC > 27 cm. A statistically significant difference was found for all measurements, confirming the arm circumference measurement as a good individual discriminator. In addition, a chi-square association test was performed to identify the relationship between an arm circumference of ≤ 27 cm and mobility restrictions. It was found that 86.8% of the older adults who cannot walk have an arm circumference ≤ 27 cm, with a statistically significant association (p <0.001).

In Table 3, the models developed from the linear regression analysis of the dependent and independent variables are presented. Five models with eight equations were developed using the arm circumference as the discriminant variable (models Ia and Ib), and with the best statistical fit (model II), best statistical adjustment stratified by sex (models IIIa and IIIb), easy-to-perform measurements (model IV), and easy-to-perform measurements stratified by sex (models Va and Vb).

**Table 1. Physical and biological characteristics of older adults by sex and age group.**

| | Population | Male | Female | 60–74 years | up to 75 years |
|---|---|---|---|---|---|
| | (n = 313) | (n = 72) | (n = 241) | (n = 67) | (n = 246) |
| | Mean (SD) | Mean (SD) | Mean (SD) | Mean (SD) | Mean (SD) |
| Age [a,b] | 82.01 (9.04) | 78.69 (9.75) | 83.00 (8.60) | 69.16 (3.82) | 85.51 (6.54) |
| Weight [a,b] | 53.64 (13.99) | 60.92 (13.22) | 51.46 (13.49) | 57.43 (14.37) | 52.60 (13.73) |
| Knee height [a] | 47.7 (3.0) | 50.7 (2.7) | 46.8 (2.5) | 48.0 (3.2) | 47.6 (3.0) |
| Waist circumference | 89.0 (13.9) | 91.2 (11.6) | 88.4 (14.5) | 89.4 (14.1) | 88.9 (13.9) |
| Arm circumference [a,b] | 24.4 (4.6) | 25.9 (3.6) | 24.0 (4.8) | 25.9 (4.7) | 24.1 (4.5) |
| Calf circumference [a] | 29.4 (5.4) | 30.8 (4.3) | 29.0 (5.6) | 30.4 (4.9) | 29.2 (5.4) |
| Subscapular skinfold [a,b] | 15.7 (8.5) | 17.6 (8.3) | 15.2 (8.4) | 18.7 (10.5) | 14.9 (7.6) |
| Triciptal skinfold | 15.7 (8.8) | 14.1 (7.8) | 16.1 (9.0) | 16.7 (10.3) | 15.4 (8.4) |

[a,b]: p-value statistically significant for Student's t-test for mean difference between males and females[a] and between age groups[b].

The analysis of the Bland-Altmann graphs showed that, in all developed equations, a cloud was formed between the standard deviations -3 and 3. No outliers (< -3 or > 3 SD) were observed for any of the equations, demonstrating that they are well adjusted in the graphical analysis of residuals with no large distortions in the observed and predicted values (Fig 1).

In Table 4, the characteristics of the development and validation groups are compared. Differences between means of the anthropometric data analyzed were not observed. The estimated weight of the developed models with the observed and estimated weights by Chumlea et al. (1988) and Rabito et al. (2008) are shown in Table 5. Note that the models developed in this study presented better ICC and respective 95% CI than those proposed by other authors.

## Discussion

Estimating the weight of the institutionalized older adults is therefore a necessity. The estimation equations in the literature have limitations regarding their anthropometric measurements,

**Table 2. Mean difference between anthropometric variables and the group of older adults with AC ≤ 27 cm and AC > 27 cm.**

| | | Mean | SD | p-value |
|---|---|---|---|---|
| Age | AC ≤ 27 cm | 83.2 | 8.7 | p < 0,001 |
| | AC > 27 cm | 78.9 | 9.2 | |
| Weight | AC ≤ 27 cm | 48.0 | 10.5 | p < 0,001 |
| | AC > 27 cm | 69.0 | 10.5 | |
| Knee height | AC ≤ 27 cm | 47.4 | 3.0 | p = 0,004 |
| | AC > 27 cm | 48.5 | 2.8 | |
| Calf circumference | AC ≤ 27 cm | 27.6 | 4.6 | p < 0,001 |
| | AC > 27 cm | 34.4 | 3.9 | |
| Waist circumference | AC ≤ 27 cm | 84.8 | 12.2 | p < 0,001 |
| | AC > 27 cm | 100.7 | 11.6 | |
| Arm circumference | AC ≤ 27 cm | 22.4 | 3.1 | p < 0,001 |
| | AC > 27 cm | 30.1 | 2.8 | |
| Subscapular skinfold | AC ≤ 27 cm | 12.8 | 6.3 | p < 0,001 |
| | AC > 27 cm | 23.6 | 8.7 | |
| Triciptal skinfold | AC ≤ 27 cm | 12.4 | 6.4 | p < 0,001 |
| | AC > 27 cm | 24.6 | 8.3 | |

**Table 3. Equations for weight estimation using anthropometric variables in older adults.**

| Model | Public | Predictive equations | R² Adjusted | SEE | Akaike |
|---|---|---|---|---|---|
| **Ia** | AC ≤ 27 cm | Weight (Kg) = (2.134 * AC) + (0.613 * CC) + (0.646 * KH) + (0.061 * AGE)–(0.797 * SEX)– 50.831 | 0.846 | 4.048 | 624.69 |
| **Ib** | AC > 27 cm | Weight (Kg) = (1.031 * AC) + (1.424 * CC) + (1.227 * KH)–(0.102 * AGE)–(1.826 * SEX)– 59.226 | 0.624 | 6.486 | 314.13 |
| **II** | Male and female | Weight (Kg) = (1.249 * CC) + (0.792 * KH) + (0.371 * WC) + (0.290 * SS)–(2.635 * SEX)– 53.556 | 0.882 | 4.816 | 972.21 |
| **IIIa** | Male | Weight (Kg) = (2.171 * CC) + (1.502 * KH) + (0.517 * TS)– 89.037 | 0.855 | 5.095 | 229.85 |
| **IIIb** | Female | Weight (Kg) = (1.255 * CC) + (0.529 * KH) + (0.358 * WC) + (0.270 * SS)– 45.338 | 0.879 | 4.635 | 726.82 |
| **IV** | Male and female | Weight (Kg) = (2.098 * CC) + (0.952 * KH)– 53.067 | 0.737 | 7.189 | 1212.14 |
| **Va** | Male | Weight (Kg) = (2.823 * CC) + (1.358 * KH)– 94.420 | 0.796 | 6.049 | 252.93 |
| **Vb** | Female | Weight (Kg) = (2.043 * CC) + (0.415 * KH)– 26.857 | 0.735 | 6.849 | 901.48 |

SEE: Standard error of estimation; AC: arm circumference; CC: calf circumference; WC: waist circumference; KH: knee height; Sex: 1 for men, 2 for women; TS: triceps skinfold; SS: subscapular skinfold.

which are difficult to obtain [10]. Moreover, they were not developed for older adults living in long-term care institutions, and therefore do not consider the particularities of the loss of physical and functional capacity. Chumlea et al. (1988) developed weight estimation equations for 228 older adults without mobility restrictions and validated them with white Americans with restricted mobility. Rabito et al. (2008) developed estimation equations for hospitalized Brazilians aged 18 years and more.

The present study aimed to develop and validate three types of weight estimation equations: a) specific ones employing a protein-energy reserve indicator as the discriminant variable, using arm circumference measurements as the discriminant (models Ia and Ib); b) more adequate equations in relation to the established statistical criteria, which used the measures that best met the agreement criteria, regardless of the variables added to the model (models II, IIIa, and IIIb); and c) equations with easy-to-perform measurements considering the older adults with mobility restrictions, which used measures such as knee height, and calf and arm circumferences (models IV, Va, and Vb). The arm circumference measurement was removed from the model because it presented collinearity with the calf circumference.

The most appropriate equation model in relation to the established statistical criteria is the one useful for estimating the weight of older adults with any condition that prevents the balance measurement but not the adequate measurement of waist circumference, triceps, and subscapular skinfold thickness.

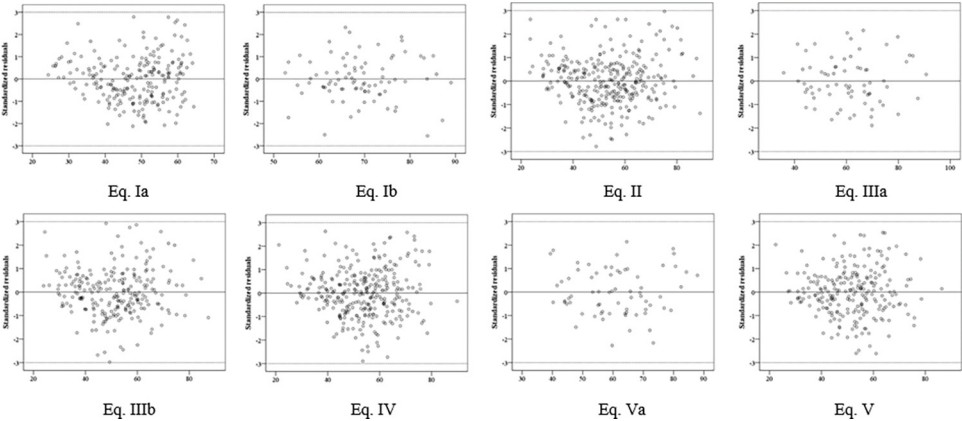

**Fig 1. Bland-Altmann plot with residues of developed equations, Brazil, 2014.**

**Table 4. Comparison of anthropometric data collected in the development and validation groups.**

| | Development group | | | Validation group | | | p-value |
|---|---|---|---|---|---|---|---|
| | n | Mean | SD | n | Mean | SD | |
| Age | 313 | 82.01 | 9.04 | 84 | 81.43 | 8.93 | 0.601 |
| Weight | 313 | 53.64 | 4.60 | 82 | 52.80 | 14.93 | 0.636 |
| Knee height | 313 | 47.69 | 3.02 | 75 | 47.80 | 3.12 | 0.776 |
| Waist circumference | 313 | 89.04 | 13.93 | 14 | 87.85 | 10.90 | 0.752 |
| Arm circumference | 313 | 24.44 | 4.60 | 78 | 23.45 | 4.81 | 0.090 |
| Calf circumference | 313 | 29.43 | 5.35 | 82 | 59.05 | 5.80 | 0.584 |
| Subscapular skinfold | 313 | 15.72 | 8.45 | 71 | 15.11 | 9.33 | 0.696 |
| Tricipital skinfold | 313 | 15.66 | 8.80 | 80 | 15.22 | 9.74 | 0.590 |

The model equations with specific estimates using the discriminant variable of the protein-energy reserve indicator were divided into two groups, an equation for the older adults with AC ≤ 27 cm and another for those with AC > 27 cm. This choice was made to make the developed models more specific. Accordingly, we have attempted to divide the study population according to the variable that most correlated with weight.

The $R^2$ of the equations differed. This can be observed in the higher value of $R^2$ in the model for older adults with AC ≤ 27 cm. It is noteworthy that this equation option, which uses an energy-protein reserve indicator as a discriminant variable, represents a differential in relation to the weight estimation equations. This is because older adults with AC ≤ 27 cm have the highest prevalence of mobility restrictions, making this model of weight estimation more accurate for this population.

In the studies by Chumlea et al. (1988) and Rabito et al. (2008), the populations did not consist of older adults with major challenges such as being bedridden or having to use wheelchairs. Rather, their participants were hospitalized adults and older adults who could walk. Therefore, in these studies, anthropometric measures were used. In practice, these measurements of older adults in long-term care institutions may yield a potential systematic error, considering that they usually cannot stand in the proper posture [10]. However, the use of anthropometric measurements in common, among the developed models and those by Chumlea et al. (1988) and Rabito et al. (2008), may explain the agreement among the models.

**Table 5. Analysis of the agreement of the estimated weight of the developed model with the observed and estimated weights using the equations proposed by Chumlea et al. (1988) and Rabito et al. (2008).**

| | Equation | n | ICC | CI ICC 95% Lower | CI ICC 95% Upper | Cronbach's Alpha |
|---|---|---|---|---|---|---|
| I | AC ≤ 27 cm | 73 | 0.855 | 0.778 | 0.906 | 0.922 |
| | AC > 27 cm | | | | | |
| II | Everyone | 10 | 0.949 | 0.809 | 0.987 | 0.974 |
| III | Male | 73 | 0.890 | 0.831 | 0.930 | 0.942 |
| | Female | | | | | |
| IV | Everyone | 73 | 0.861 | 0.787 | 0.910 | 0.925 |
| V | Male | 73 | 0.842 | 0.760 | 0.898 | 0.914 |
| | Female | | | | | |
| | Chumlea et al. 1988 | 64 | 0.918 | 0.869 | 0.950 | 0.957 |
| | Rabito et al. 2008 | 14 | 0.923 | 0.779 | 0.975 | 0.960 |

ICC: Intraclass Correlation Coefficient; CI 95%: Confidence Interval of the ICC; AC: arm circumference.

For older adults living in long-term care institutions, Lima et al. (2016) considered the following measures as difficult to obtain: a) Half arm span and demi-span: used in models of height estimation, which are hampered by joint stiffness and generate problems for the older adults in terms of standing in the proper position. This does not allow sufficient time to take the measurement. b) Waist circumference: used in weight estimation models, it is measured with the older adult standing. This is impossible to do safely and following the proper technique for those who are bedridden or in wheelchairs. c) Subscapular skinfold: used in weight estimation models. This is performed in the posterior coronal plane, which is in contact with a chair or bed.

Therefore, for the development of the models, waist circumference and subscapular skinfold were excluded, since the intention was to develop a viable estimation equation for the evaluation of older adults with restricted mobility, especially for those using wheelchairs or who are bedridden. Although the equations developed with waist circumference and subscapular skinfold values may have higher $R^2$ values and would therefore be closer to the observed weight, there is no proper technique to measure waist circumference and subscapular skinfold.

Equations VI, Va, and Vb employ measurements that only require an anthropometric tape and anthropometric ruler to be obtained, which can be acquired at a low cost by LTCIs. In addition, these measures can be taken with older adults lying in bed or sitting in a wheelchair without major difficulties and bias from the proper technique.

Regarding the criteria for taking the measures, the equation satisfies the statistical criteria as it presents considerable $R^2$ values. Here, the ideal is that the estimation model: (a) applies to the public most likely to have mobility restrictions and (b) uses measures that can be taken despite physical restrictions. The equations developed in this study have considered these criteria, as it would not be useful to formulate statistically fit equations that cannot be applied in clinical practice.

The choice of anthropometric measurements included in equations IV, Va, and Vb was based on the feasibility of the estimate. Using the equations developed for weight estimation provides a good estimate by taking two simple anthropometric measurements (calf circumference and knee height), which can be performed using low-cost equipment (an anthropometric tape and a ruler).

Weight measurement of older adults living in long-term care institutions is important since the assessment of nutritional status is essential in monitoring healthy aging. Weight is a measure used to calculate the body mass index (BMI) and assess any variation in weight percentage [14].

Alternative methods for weight measurement are common, such as weighing the older adults in the arm of a caregiver and later weighing the caregiver to calculate the difference. However, this method compromises the ergonomics of the caregiver and endangers the physical integrity of older adults. An alternative for weight measurement is the equipment used in this study (SECA 985Ⓡ). However, considering that, in Brazil, it costs the equivalent of 31.9 minimum salaries in 2021, it will not be easily purchased by LTCIs in the country.

Obtaining the weight estimate provides the LTCIs with the data required to monitor the weight of older adults [8]. The best indicator of anthropometric nutritional status is BMI, which is calculated using the measures of weight and height [1]. If older adults have mobility restrictions that justify the estimation of weight, it is likely that their height will also need to be estimated, or that the information provided about individuals will be used.

## Conclusion

A set of five equations for estimating weight were developed and validated. Three groups of differentiated weight estimation equations were developed and validated as follows: a) specific equations employing the discriminant variable of a protein-energy reserve indicator (AC $\leq$ 27 cm), b)

more appropriate equations in relation to the established statistical criteria, and c) equations with easy-to-perform measurements considering older adults with mobility restrictions. In the first and last groups, generalized equations were developed and stratified by sex.

The choice of which equation to use should consider the physical capacity of the older adult to be evaluated. It is recommended to use the most appropriate equation in relation to the established statistical criteria. If this is not possible, the other developed equations should be used.

Those who need a weight estimation method will need an estimate of height as well, to calculate BMI and classify the nutritional status. However, the weight alone is useful in terms of monitoring nutritional interventions and prescribing medication.

## Author Contributions

**Conceptualization:** Marcos Felipe Silva de Lima, Natália Louise de Araújo Cabral, Larissa Praça de Oliveira, Laura Camila Pereira Liberalino, Kenio Costa de Lima, Clélia de Oliveira Lyra.

**Data curation:** Marcos Felipe Silva de Lima, Natália Louise de Araújo Cabral, Severina Carla Vieira Cunha Lima, Kenio Costa de Lima, Clélia de Oliveira Lyra.

**Formal analysis:** Marcos Felipe Silva de Lima, Severina Carla Vieira Cunha Lima, Kenio Costa de Lima, Clélia de Oliveira Lyra.

**Funding acquisition:** Larissa Praça de Oliveira, Lucia de Fátima Campos Pedrosa, Kenio Costa de Lima, Clélia de Oliveira Lyra.

**Investigation:** Marcos Felipe Silva de Lima, Laura Camila Pereira Liberalino, Clélia de Oliveira Lyra.

**Methodology:** Marcos Felipe Silva de Lima, Natália Louise de Araújo Cabral, Clélia de Oliveira Lyra.

**Project administration:** Larissa Praça de Oliveira, Lucia de Fátima Campos Pedrosa, Kenio Costa de Lima, Clélia de Oliveira Lyra.

**Resources:** Marcos Felipe Silva de Lima, Severina Carla Vieira Cunha Lima, Clélia de Oliveira Lyra.

**Software:** Marcos Felipe Silva de Lima, Clélia de Oliveira Lyra.

**Supervision:** Marcos Felipe Silva de Lima, Kenio Costa de Lima, Clélia de Oliveira Lyra.

**Validation:** Marcos Felipe Silva de Lima, Kenio Costa de Lima, Clélia de Oliveira Lyra.

**Visualization:** Marcos Felipe Silva de Lima, Clélia de Oliveira Lyra.

**Writing – original draft:** Marcos Felipe Silva de Lima, Clélia de Oliveira Lyra.

**Writing – review & editing:** Marcos Felipe Silva de Lima, Clélia de Oliveira Lyra.

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
