## [Decision Letter · Decision Letter 0]

27 Oct 2021

PONE-D-21-13956DEVELOPMENT AND VALIDATION OF PREDICTIVE WORKABLE WEIGHT EQUATIONS FOR ELDERLY RESIDENTS IN LONG-TERM CARE INSTITUTIONSPLOS ONE

Dear Dr. Lima,

Thank you for submitting your manuscript to PLOS ONE. After careful consideration, we feel that it has merit but does not fully meet PLOS ONE’s publication criteria as it currently stands. Therefore, we invite you to submit a revised version of the manuscript that addresses the points raised during the review process.

The manuscript has been evaluated by two reviewers, and their comments are available below.

The reviewers have raised a number of concerns that need attention. They request additional information on methodological aspects of the study (such as details of the study setting and statistical methods), and revisions to analyses.

Could you please revise the manuscript to carefully address the concerns raised?

We look forward to receiving your revised manuscript.

Kind regards,

Marianne Clemence

Associate Editor

PLOS ONE

Journal Requirements:

4. Please ensure that you include a title page within your main document. You should list all authors and all affiliations as per our author instructions and clearly indicate the corresponding author.

Reviewers' comments:

Reviewer's Responses to Questions

**Comments to the Author**

1. Is the manuscript technically sound, and do the data support the conclusions?

Reviewer #1: Partly

Reviewer #2: Yes

2. Has the statistical analysis been performed appropriately and rigorously? 

Reviewer #1: Yes

Reviewer #2: Yes

3. Have the authors made all data underlying the findings in their manuscript fully available?

Reviewer #1: No

Reviewer #2: Yes

4. Is the manuscript presented in an intelligible fashion and written in standard English?

Reviewer #1: No

Reviewer #2: Yes

5. Review Comments to the Author

Reviewer #1: The study is really necessary study since the authors address a very important aspect of health of older people. I have few concerns with this study.

1. Better to exclude the 55 individuals who participated for the study at 1st time from the 2nd stage and re analyze the data.

2. There should be a coherence in the writing specially the methods, data analysis, results and discussion.

3. Methods – study design, period of data collection, stage I and 2, way of taking anthropometry information, data analysis and ethical clearance.

4. Data analysis – how did you present the descriptive data, removing outliers, comparing the characteristics of two group (development and validation groups; there should not be difference), stratification, correlation analysis (between weight and anthropometry), regression analysis to choose the model/equations, Bland-Altman plots, comparing with previous study equations etc.

5. Results should be arranged in the same sequence of results analysis.

6. Discussion – summary of findings, reasons for observations, similarities and discrepancies with the existing literature, reasons for similarities and discrepancies, applications of these findings, strengths and limitation, recommendations.

7. Need language improvement.

Reviewer #2: This is a relevant topic since in older adults’ weight is the anthropometric measure that best relates to malnutrition, especially if there are changes from baseline weight. Furthermore, weight must be known in order to use tools such as body mass index (BMI) and the Mini Nutritional Assessment test (MNA) for nutritional screening. However, in hospitalized or institutionalized older adults, it is not always possible to determine weight on a conventional scale. Therefore, it is important to find equations that could serve to perform screening for malnutrition.

The goal of the work is clearly described.

The work presented is ethically sound. Authors have submitted the research protocol for approval by an University ethics committee.

Older individuals become more heterogeneous with age, a specific descriptor such as elderly is inaccurate and misleading thus authors should consider the replacement of “elderly” by “older adults” in all the manuscript .

Title: The title should mention “older adults”

Introduction: In the 3th paragraph last sentence “As such,……” should be rewritten

Methods: Inclusion and exclusion criteria of the patients should be presented

Results

Table2- Authors should correct the legend in order to consider the arm circumference (AC)

In the last paragraph of the discussion the abbreviations ICC and CI should be described: (Intraclass Correlation Coefficient (ICC)………..)

Discussion: Authors might hypothesise about the concordance of their models with those of Chumlea et al. (1988) and Rabito et al. (2008) .

Considering the above comments, I recommend that manuscript could be accepted with minor revisions

6. PLOS authors have the option to publish the peer review history of their article (what does this mean?). If published, this will include your full peer review and any attached files.

Reviewer #1: No

Reviewer #2: No

---

## [Author Response · Author response to Decision Letter 0]

15 Feb 2022

Reviewer #1: The study is really necessary study since the authors address a very important aspect of health of older people. I have few concerns with this study.

1. Better to exclude the 55 individuals who participated for the study at 1st time from the 2nd stage and re analyze the data.

A: Done. Results did not change.

2. There should be a coherence in the writing specially the methods, data analysis, results and discussion.

A: Sections were adjusted.

3. Methods – study design, period of data collection, stage I and 2, way of taking anthropometry information, data analysis and ethical clearance.

A: Sections were adjusted.

4. Data analysis – how did you present the descriptive data, removing outliers, comparing the characteristics of two group (development and validation groups; there should not be difference), stratification, correlation analysis (between weight and anthropometry), regression analysis to choose the model/equations, Bland-Altman plots, comparing with previous study equations etc.

A: Sections were adjusted.

5. Results should be arranged in the same sequence of results analysis.

A: Done. Sections were adjusted.

6. Discussion – summary of findings, reasons for observations, similarities and discrepancies with the existing literature, reasons for similarities and discrepancies, applications of these findings, strengths and limitation, recommendations.

A: Section was adjusted.

7. Need language improvement.

A: Language improvement was done.

Reviewer #2: This is a relevant topic since in older adults’ weight is the anthropometric measure that best relates to malnutrition, especially if there are changes from baseline weight. Furthermore, weight must be known in order to use tools such as body mass index (BMI) and the Mini Nutritional Assessment test (MNA) for nutritional screening. However, in hospitalized or institutionalized older adults, it is not always possible to determine weight on a conventional scale. Therefore, it is important to find equations that could serve to perform screening for malnutrition.

The goal of the work is clearly described.

The work presented is ethically sound. Authors have submitted the research protocol for approval by an University ethics committee.

Older individuals become more heterogeneous with age, a specific descriptor such as elderly is inaccurate and misleading thus authors should consider the replacement of “elderly” by “older adults” in all the manuscript .

A: Done. It was replaced “elderly” by “older adults” in all the manuscript.

Title: The title should mention “older adults”

A: Done. Title has the term “older adults”.

Introduction: In the 3th paragraph last sentence “As such,……” should be rewritten

A: Done. 

Methods: Inclusion and exclusion criteria of the patients should be presented

A: Done.

Results

Table2- Authors should correct the legend in order to consider the arm circumference (AC)

A: Done. Thanks for attention.

In the last paragraph of the discussion the abbreviations ICC and CI should be described: (Intraclass Correlation Coefficient (ICC)………..)

A: Done. 

Discussion: Authors might hypothesise about the concordance of their models with those of Chumlea et al. (1988) and Rabito et al. (2008) .

A: Done. 

Considering the above comments, I recommend that manuscript could be accepted with minor revisions

---

## [Decision Letter · Decision Letter 1]

5 Aug 2022

PONE-D-21-13956R1Development and validation of predictive workable weight equations for older adults brazillian Long-Term Care residentsPLOS ONE

Dear Dr. Lima,

Thank you for submitting your manuscript to PLOS ONE. After careful consideration, we feel that it has merit but does not fully meet PLOS ONE’s publication criteria as it currently stands. Therefore, we invite you to submit a revised version of the manuscript that addresses the points raised during the review process.

Specifically, the reviewer still has a could pf concerns including language usage. Please note that PLOS ONE does not provide copyediting or proofs of accepted manuscripts. We therefore recommend that you carefully review your manuscript and correct any errors at this time.

We look forward to receiving your revised manuscript.

Kind regards,

Jianhong Zhou

Staff Editor

PLOS ONE

Journal Requirements:

Reviewers' comments:

Reviewer's Responses to Questions

**Comments to the Author**

1. If the authors have adequately addressed your comments raised in a previous round of review and you feel that this manuscript is now acceptable for publication, you may indicate that here to bypass the “Comments to the Author” section, enter your conflict of interest statement in the “Confidential to Editor” section, and submit your "Accept" recommendation.

Reviewer #2: All comments have been addressed

Reviewer #3: (No Response)

2. Is the manuscript technically sound, and do the data support the conclusions?

Reviewer #2: Yes

Reviewer #3: Yes

3. Has the statistical analysis been performed appropriately and rigorously? 

Reviewer #2: Yes

Reviewer #3: Yes

4. Have the authors made all data underlying the findings in their manuscript fully available?

Reviewer #2: Yes

Reviewer #3: Yes

5. Is the manuscript presented in an intelligible fashion and written in standard English?

Reviewer #2: Yes

Reviewer #3: Yes

6. Review Comments to the Author

Reviewer #2: Authors have respond to all the questions adressed. Moreover they have improved the language in order to accomplish the concerns of the reviewers. Considering the above manuscript should be accepted.

Reviewer #3: The manuscript presents an interesting topic, with relevance for clinical practice.

However, there are still issues that need to be clarified:

- Methods: please provide the range of values for the technical error of measurement.

- Results: please explain the sentence: "The characterization of the study population showed that the anthropometric measurements taken differed in the groups studied".

- The conclusion should contain an objective response to the aim of the study.

- A general review of the English language is suggested throughout the text.

7. PLOS authors have the option to publish the peer review history of their article (what does this mean?). If published, this will include your full peer review and any attached files.

Reviewer #2: No

Reviewer #3: No

---

## [Author Response · Author response to Decision Letter 1]

28 Sep 2022

. Review Comments to the Author

Reviewer #2: Authors have respond to all the questions adressed. Moreover they have improved the language in order to accomplish the concerns of the reviewers. Considering the above manuscript should be accepted.

Reviewer #3: The manuscript presents an interesting topic, with relevance for clinical practice.

However, there are still issues that need to be clarified:

- Methods: please provide the range of values for the technical error of measurement.

Done.

- Results: please explain the sentence: "The characterization of the study population showed that the anthropometric measurements taken differed in the groups studied".

The sentence was changed for “The characterization of the study population showed that the anthropometric measurements were different for each studied group”.

- The conclusion should contain an objective response to the aim of the study.

Done. It was added to conclusion “A set of five equations for estimating weight were developed and validated”.

- A general review of the English language is suggested throughout the text.

Done. Language was improved by translate review.

---

## [Decision Letter · Decision Letter 2]

2 Nov 2022

PONE-D-21-13956R2Development and validation of predictive workable weight equations for older adults brazillian Long-Term Care residentsPLOS ONE

Dear Dr. Lima,

Thank you for submitting your manuscript to PLOS ONE. After careful consideration, we feel that it has merit but does not fully meet PLOS ONE’s publication criteria as it currently stands. Therefore, we invite you to submit a revised version of the manuscript that addresses the points raised during the review process.

 Specifically, please specify the range of values for the technical error of measurement as requested by the reviewer.

We look forward to receiving your revised manuscript.

Kind regards,

Jianhong Zhou

Staff Editor

PLOS ONE

Journal Requirements:

Reviewers' comments:

Reviewer's Responses to Questions

**Comments to the Author**

1. If the authors have adequately addressed your comments raised in a previous round of review and you feel that this manuscript is now acceptable for publication, you may indicate that here to bypass the “Comments to the Author” section, enter your conflict of interest statement in the “Confidential to Editor” section, and submit your "Accept" recommendation.

Reviewer #3: (No Response)

2. Is the manuscript technically sound, and do the data support the conclusions?

Reviewer #3: Yes

3. Has the statistical analysis been performed appropriately and rigorously? 

Reviewer #3: Yes

4. Have the authors made all data underlying the findings in their manuscript fully available?

Reviewer #3: Yes

5. Is the manuscript presented in an intelligible fashion and written in standard English?

Reviewer #3: Yes

6. Review Comments to the Author

Reviewer #3: The authors addressed the comments raised in the previous review, except the following comment: "Please provide the range of values for the technical error of measurement".

This reviewer could not find the values in the manuscript text.

7. PLOS authors have the option to publish the peer review history of their article (what does this mean?). If published, this will include your full peer review and any attached files.

Reviewer #3: No

---

## [Author Response · Author response to Decision Letter 2]

16 Nov 2022

We've checked your submission and before we can proceed, we need you to address the following issues:

1. Please amend the title either on the online submission form or in your so that they are identical. 

Title changed on the online submission form for “Development and validation of predictive workable weight equations for Brazilian older adult residents in Long-Term Care Institutions” according to manuscript.

---

## [Editor Report · Decision Letter 3]

4 Jan 2023

Development and validation of predictive workable weight equations for Brazilian older adult residents in Long-Term Care Institutions

PONE-D-21-13956R3

Dear Dr. Lima,

We’re pleased to inform you that your manuscript has been judged scientifically suitable for publication and will be formally accepted for publication once it meets all outstanding technical requirements.

Kind regards,

Jianhong Zhou

Staff Editor

PLOS ONE
---

## [Editor Report · Acceptance letter]

8 Jan 2023

PONE-D-21-13956R3 

Development and validation of predictive workable weight equations for Brazilian older adult residents in Long-Term Care Institutions 

Dear Dr. Lima:

I'm pleased to inform you that your manuscript has been deemed suitable for publication in PLOS ONE. Congratulations! Your manuscript is now with our production department. 

Kind regards, 

on behalf of

Jianhong Zhou 

Staff Editor

PLOS ONE